# Polysubstance use among adolescents in Malaysia: Findings from the National Health and Morbidity Survey 2017

Wan Shakira Rodzlan Hasani[1]*, Thamil Arasu Saminathan[1], Nur Liana Ab Majid[1], Jane Ling Miaw Yn[1], Halizah Mat Rifin[1], Hamizatul Akmal Abd Hamid[1], Tania Gayle Robert Lourdes[1], Ahzairin Ahmad[1], Hasimah Ismail[1], Rusdi Abd Rashid[2], Muhammad Fadhli Mohd Yusoff[1]

1 Institute for Public Health, National Institutes of Health, Ministry of Health Malaysia, Setia Alam, Selangor, Malaysia, 2 Department of Psychological Medicine, Faculty of Medicine, Center for Addiction Science Studies (UMCAS), University of Malaya, Kuala Lumpur, Malaysia

* shaki_iera@yahoo.com

**Data Availability Statement:** The National Institute of Health, Ministry of Health Malaysia (data ethics committee) has placed restriction on sharing the

## Abstract

### Background

Polysubstance use is defined as the use of more than one non-prescribed licit or illicit substance either concurrently or simultaneously. This study aimed to determine the prevalence of single substance users and polysubstance users and 'their associated factors among adolescents in Malaysia.

### Methods

This study was a secondary data analysis from the National Health and Morbidity Survey (NHMS) 2017, a cross-sectional survey conducted among Malaysian school-going adolescents aged 13 to 17. The NHMS utilised a two-stage stratified cluster sampling. Multivariate Multinomial Logistic Regression analysis was applied.

### Results

The overall prevalence of single substance use and polysubstance use among adolescents were 17.2% and 5.1% respectively. The multinomial model showed a higher likelihood of being single or polysubstance user among male (single user OR = 3.0, poly user OR = 4.6), others Bumiputeras vs Malay (single user OR = 1.7, poly user OR = 5.3), those who live with a single parent (single user OR = 1.2, poly user OR = 1.4), involved in truancy (single user OR = 1.7, poly user OR = 3.6) and being bullied (single user OR = 1.3, poly user OR = 3.4), those who had lack of peer support (single user OR = 1.3, poly user OR = 1.4), poor parental bonding (single user OR = 1.4, poly user OR = 1.8), depression (single user OR = 1.4, poly user OR = 3.2) and those who had no close friend (single user OR = 1.3, poly user OR = 2.7).

full dataset due to cases involving researchers manipulating the data. Interested researchers will need to send a formal letter/email to the Director General of Health Malaysia, together with the data request form and proposal, available at (http://iku. moh.gov.my/images/IKU/Document/Form/ Borangpermohonandatalatest.pdf). The proposal will be reviewed by the Data Repository team from Biostatistics Sector, National Institute of Health Malaysia to ensure no duplication with other projects that have used the NHMS 2017 data. The data request flow chart is available on the website and can be accessed here (http://iku.moh.gov.my/ images/IKU/Document/Form/ FlowChartforIPHDataApplication.pdf). The authors also confirm they did not have any special access privileges that others would not have for the NHMS 2017 data. The contact information of the Director General of Health Malaysia as below; Address: Director General's Office, Ministry of Health Malaysia, Kompleks E, Aras 12, Blok E7, Presint 1, 62590 Putrajaya, Malaysia Email: anhisham@moh. gov.my Contact number: +60388832545, Fax number: +60388895542

**Funding:** Funded studies National Institute of Health, Ministry of Health Malaysia.

**Competing interests:** The authors have declared that no competing interests exist.

## Conclusion

Our study highlighted multiple significant associated factors of single and polysubstance use among adolescents in Malaysia. This result can assist in the development of specific intervention and prevention programs targeting high-risk groups.

## Introduction

Polysubstance use among adolescents is a public health concern and remains an essential topic for practitioners and policymakers. Polysubstance use is defined as using more than one non-prescribed licit or illicit psychoactive substance either concurrently or simultaneously [1,2]. Licit psychoactive substances include alcohol and cigarette, while illicit psychoactive substances include marijuana, cocaine, heroin, lysergic diethylamide (LSD), and amphetamines [3]. The use of licit and illicit substances frequently starts among schoolchildren during adolescents. Adolescence is a period associated with the highest risk for initiation of polysubstance use, with tobacco and alcohol use usually preceding illicit drugs. Another study found that alcohol, cocaine, and marijuana were the most commonly abused substances, both alone and in combination [4]. In Southeast Asian countries, alcohol and tobacco are commonly used polysubstance among adolescents; the Philippines (10.2%) had the highest prevalence followed by Thailand (7.4%) and Malaysia (2.7%) [5]. In Malaysia, there were increased use of alcohol (8.9% to 10.2%), tobacco (11.5% to 13.8%), and also illicit drug (1.5% to 3.4%) among adolescents from the year 2012 to 2017 [6,7]. The increased availability of alcohol and cigarette in Malaysia and lack of enforcement at ground level on under-age purchases of these substances makes adolescent more susceptible to early initiation of such substances. It has been reported that cigarettes in Malaysia are easily accessible by adolescents [8]. Alcohol drinks are also readily available in a coffee shop, supermarket and sundry shop.

Polysubstance use of alcohol, drug or tobacco, in any combination, were associated with more significant morbidity and mortality risk compared to use of a single substance only [9]. Polysubstance use could increase the likelihood of physical or psychological harm [10]. The previous study on adolescents demonstrated the link between polysubstance use with negative outcomes, including smoking in adulthood [11], risky sexual behaviour [12], and substance dependence [13]. Polysubstance uses were also associated with an increased risk of adverse psychological experiences, including depression and negative social consequences [14–16]. Other than socio-economic and physical environment factors, previous literature also reported other significant factors that contributed to substance use, such as unsupportive family condition, patient-child relationship, and being raised by a single parent [17,18]. Other factors, such as loneliness, had no close friends, inadequate peer support, involved in truancy, and being bullied, can also render adolescents vulnerable to substance use [19].

Adolescence is a period of vulnerability for developing polysubstance use. Adolescents may benefit from targeted risk profiling for prevention measures. However, there has been a lack of research exploring polysubstance use in Malaysia, especially among the younger age population. Thus, this study aims to determine the prevalence of single substance user and polysubstance user and their associated factors among adolescents in Malaysia.

## Material and method

### Participants

For this study, the data was extracted from the National Health and Morbidity Survey (NHMS) 2017: Adolescents Health Survey. This survey was a cross-sectional study targeted at

school-going adolescents in Malaysia aged 13 to 17. NHMS 2017 aimed to provide data on health risk behaviours and protective factors among adolescents. A two-stage stratified cluster sampling was applied to ensure the representativeness of the samples. The first stage involves selecting schools with stratification by state, and the second stage consists of the selection of the classes at the schools chosen. All students within the selected classes were recruited as respondents. The selection of schools within the state was performed using systematic probability sampling proportionate to school enrolment size, while the selection of the classes was by systematic random sampling. Overall, a total of 212 secondary schools were selected to participate in this survey.

The sample size was calculated using a single proportion formula to estimate prevalence with adjustment for the total number of the target population, design effect, and non-response rate of 25%. Thus, the optimum sample size required was 30, 496 respondents. In this dataset, interviews were successfully conducted on 27,497 school-going adolescents from the 212 selected schools in Malaysia. The detailed methodology of NHMS 2017 was described and published by Awaluddin et al. [20].

## Measures

Data on substance use was collected using a validated self-administered bilingual questionnaire adopted from the Malaysian Global School Health Survey (GSHS) 2012 [21]. The GSHS used a standardised questionnaire consisting of several modules to gather information on health risk behaviours such as alcohol, tobacco, and drug use, and protective factors that are compulsory for all countries participating in the GSHS [21]. The optical answer sheets were used, and the answer sheets were anonymous to ensure student confidentiality.

**Dependent variables.**   Substance use for this study refers to the use of alcohol, cigarettes, and illicit drugs. Illicit drugs include heroin, morphine, glue, amphetamine, methamphetamines (ecstasy, syabu, ice), and marijuana/cannabis (except prescribed medicine). The outcome variable for this analysis is "substance use," which was coded into three groups; "non-user = 0", "single substance user = 1," and "polysubstance user = 2". This study defined "current single substance user" as "had used any substance at least once in the past 30 days" while "polysubstance user" is defined as "concurrent use of more than one substance in the past 30 days".

"Current user of the illicit drug" was measured using the following question; "in the past 30 days, how many times have you used drugs?" with response options from "0 times" to "20 or more times". Those who responded other than "0 times" were classified as "current user for an illicit drug."

"Current smoking" was assessed using the following items: (1) "During the past 30 days, on how many days did you smoke a cigarette?" with response options range from "0 days" to "all 30 days", (2) "During the past 30 days, did you use either shisha, traditional hand-rolled cigarette, cigar or pipe smoking?" (Choice of response: "yes" or "no"). Respondents who answered other than "0 days" for question (1), and/ or "yes" to either tobacco products in question (2), were classified as "current smokers."

"Current alcohol user" was measured using the following item: "during the past 30 days, on how many days did you have at least one drink containing alcohol?" with response options range from "0 days" to all "30 days". Respondents who answered other than "0 days" were classified as "current drinkers."

The lifetime use of illicit drugs, tobacco, and alcohol was also presented in this study to describe the overall prevalence of substance use in Malaysia. "Lifetime user of substance" was

defined as "ever used any smoked tobacco product, ever drank alcohol or ever used illicit drug in the lifetime."

**Independent variables.**   The independent variables used for analysis were socio-demographic, protective factors, mental health problems, truancy, and being bullied. Demographic variables included were gender (male, female), age group (13–15, 16–17 years), school's location (urban, rural), ethnicity (Malay, Chinese, Indian, other Bumiputeras), and marital status of parents (married, separated).

The following questions were asked for the protective factors variables during the past 30 days prior to the study: 1) Question for peer support "how often most of the students in your school were kind and helpful?", 2) Question for parental bonding "how often did your parents or guardians really know what you were doing during your free time?", 3) Question for parental respect for privacy "how often did your parents or guardians go through your personal things without your approval?". All the questions were assessed by five responses ranging from "(1) = Never", "(2) = rarely", "(3) = sometimes", "(4) = most of the time" and "(5) = always". Option (1) to (3) were classified as "inadequate" whereas options (4) to (5) as "adequate" for peer support and parental bonding while option "never" or "rarely" were classified as parents to "have respect for privacy."

Data on truancy was obtained using this question "During the past 30 days, on how many days did you miss classes or school without permission?". This question's response options range from "0 days" to "10 or more days". Respondents who answered other than "0 days" were classified as "yes" for truancy. For bullied experience, "during the past 30 days, on how many days were you bullied? (Options: 0 days to all 30 days). Those who answered other than "0 days" were classified as "being bullied."

Mental health problems such as anxiety, depression, and stress were measured using self-administered bilingual Depression Anxiety Stress Scale (DASS-21) questionnaires, which have been translated and validated for the Malaysian population [22]. Other mental health problems such as "loneliness" and "had no close friend" were measured using the standard questionnaire from GSHS. Those who responded either "most of the time" or "always" for feeling lonely during the past 12 months before the study was defined as "feeling lonely most of the time or always" while those who responded "do not have any close friend" was classified as "no close friend."

## Data analysis

The self-administered data from Optical Mark Recognition (OMR) form was captured by the scanner in Excel form and exported to SPSS statistical software version 22 for analysis. Data on substance use and selected variables were extracted from the NHMS's data. A complex sampling descriptive analysis was used to estimate the prevalence of single substance use and polysubstance use among adolescents in Malaysia. The associated factors for single substance use and polysubstance use compared to non-user were identified using Multinomial Logistic Regression analysis using STATA software version 14. A Wald test was used to assess each variable's contribution to the model. In this study, a univariable analysis was carried out by testing all the 15 potential predictor variables to screen for important independent variables. Considering the variables with $p$-values <0.25 from univariable analysis and important variables based on biological plausibility, all those 15 variables were included in the preliminary final model (variable selection). The variable selection is the process of "reducing the model" to get the best fit model by including all the candidate variables in the model and repeatedly removing the variables with the highest non-significant p-value until the model contains only significant terms. Hence, the final model was created based on ten variables significantly associated

at the level of p <0.05 during the final steps of variables selection. Those variables were age, sex, ethnicity, marital status of parents, truancy, peer support, parent/guardian bonding, depression, a close friend, and being bullied. Multicollinearity and interaction were checked accordingly. The overall fitness was checked using a weighted classification table and weighted ROC (receiver operating characteristic) curve for each binary logit model. The findings were presented as crude and adjusted odds ratios with their 95% confidence intervals. The analyses utilised the complex sampling design to account for the sample weightage and study design properties, including design effect, stratification, and clustering in the sample design.

### Ethical approval and field implementation

This study obtained ethical approval from the Medical Research and Ethics Committee (MREC), Ministry of Health Malaysia, and also approval from the Education Planning and Research Division, Ministry of Education Malaysia. Prior to the survey, several meetings with the selected schools' focal point were conducted to inform them about the survey and brief about the parents' consent form. The consent forms were distributed a week before the survey. Students and their parents' consent forms were collected just before the actual day of the survey. Students who refused to participate or did not submit parental/ guardian consent were considered as non-response of eligible participants in this survey. Before the survey, the data collectors were trained during a 1-week workshop. The data collection was conducted from March until May 2017. A field supervisor was assigned for each state to supervise the survey activities, including arranging the slot for the survey with the school. The survey was conducted either in the classroom or school hall during school time, and it took about 45 minutes to 1 hour to complete the survey. The answered OMR form was put in the envelope by each student to maintain confidentiality.

## Results

The demographic characteristics of survey respondents are shown in Table 1. The total number of respondents was 27,497, with an overall response rate of 89.0%. There were an almost

**Table 1. Socio-demographic characteristic of school-going adolescents in Malaysia (n = 27497).**

| Socio-Demographic | n (%) |
|---|---|
| **Gender** | |
| Male | 13135 (47.8) |
| Female | 14362 (52.2) |
| **Age group** | |
| 13–15 years | 17042 (62.0) |
| 16–17 years | 10455 (38.0) |
| **Locality of School** | |
| Urban | 15899 (57.8) |
| Rural | 11598 (42.2) |
| **Ethnic** | |
| Malay | 18713 (68.1) |
| Chinese | 4100 (14.9) |
| Indian | 1428 (5.2) |
| Other Bumiputeras | 3256 (11.8) |
| **Marital Status of Parents** | |
| Married and living together | 22629 (84.0) |
| Divorced/Widowed/Separated | 4302 (16.0) |

**Table 2. The lifetime and current prevalence of substance use among school-going adolescents in Malaysia.**

| Substance use | Total | Male | Female |
|---|---|---|---|
| | % (95% CI) | % (95% CI) | % (95% CI) |
| **Smoked Tobacco** | | | |
| Lifetime use | 21.4 (20.04, 22.84) | 34.3 (32.09, 36.55) | 8.7 (7.71, 9.88) |
| Current use | 15.9 (14.72, 17.26) | 25.3 (23.28, 27.50) | 6.7 (5.83, 7.71) |
| **Alcohol** | | | |
| Lifetime use | 19.3 (17.10, 21.70) | 21.8 (19.30, 24.50) | 16.8 (14.40, 19.50) |
| Current use | 10.2 (9.00, 11.60) | 12.8 (11.30, 14.50) | 7.7 (6.50, 9.10) |
| **Any Illicit drug use** | | | |
| Lifetime use | 4.3 (3.64, 5.08) | 6.6 (5.60, 7.80) | 2.0 (1.57, 2.63) |
| Current use | 3.4 (2.83, 4.12) | 5.3 (4.47, 6.37) | 1.5 (1.10, 2.12) |

The analysis using complex sample design method (weighted analysis)

equal proportion of male (47.8%) and female (52.2%) respondents, as well for urban (57.8%) and rural (42.2%) area. More than half of the respondents were Malay (68.1%), aged 13–15 years (62.0%), and had parents who were married/living together (84.0%).

The prevalence of substance use is shown in Table 2. The most common current substance use was tobacco (15.9%). The overall prevalence of single and polysubstance use was 17.2% (95% CI: 16.30, 18.20) and 5.1% (95% CI: 4.34, 5.96), respectively (Table 3). In general, substance use was significantly higher among males than females. For adolescent males, the most commonly used substances were smoked tobacco while for females, alcohol is the most commonly used substance. The combination of tobacco and alcohol is the most common pair of substance use among adolescents in Malaysia [(2.0 (1.66, 2.51)].

The multinomial model showed that males were more likely to be substance users than females regardless of whether it was single or polysubstance use (single user OR = 3.0, poly user OR = 4.6). Those of younger age (13–15 years) had higher odds of single substance use than those of older age. However, age was not significant for polysubstance use (Table 4).

This study also showed a higher likelihood of single or polysubstance use among other Bumiputeras (single user OR = 1.7, poly user OR = 5.3), Indian (poly user OR = 4.4), and

**Table 3. Prevalence of single substance use and polysubstance use among school-going adolescents in Malaysia.**

| Substance use | Total | Male | Female |
|---|---|---|---|
| | % (95% CI) | % (95% CI) | % (95% CI) |
| **Single Substance user**[*] | 17.2 (16.30, 18.20) | 24.1 (22.71, 25.64) | 10.4 (9.36, 11.56 |
| Illicit drug only | 0.4 (0.25, 0.58) | 0.5 (0.30, 0.78) | 0.3 (0.12, 0.64) |
| Tobacco only | 11.2 (10.34, 12.15) | 17.9 (16.33, 19.66) | 4.6 (4.07, 5.18) |
| Alcohol only | 5.6 (4.82, 6.58) | 5.7 (4.76, 6.88) | 5.5 (4.66, 6.58) |
| **Polysubstance user**[**] | 5.1 (4.34, 5.96) | 7.9 (6.76, 9.26) | 2.3 (1.78, 2.96) |
| Illicit drug and tobacco | 2.7 (2.16, 3.32) | 4.3 (3.53, 5.28) | 1.1 (0.74, 1.52) |
| Illicit drug and alcohol | 2.5 (2.03, 3.19) | 4.0 (3.22, 5.08) | 1.1 (0.78, 1.50) |
| Tobacco and alcohol | 4.2 (3.55, 5.05) | 6.6 (5.44, 7.88) | 1.9 (1.49, 2.54) |
| illicit drug, alcohol and tobacco use | 2.2 (1.71, 2.79) | 3.5 (2.74, 4.45) | 0.9 (0.63, 1.28) |

The analysis using complex sample design method (weighted analysis)

[*]currently use either one substance at one time

[**]concurrently use more than one any type of substance

**Table 4. Factor associated with single and polysubstance use among Malaysian adolescent from Univariate & Multivariate Multinomial Logistic Regression analysis (n = 25, 975).**

| Predictive factors | Complex Sample Univariate Multinomial logistic regression (Crude Odd Ratio, OR) | | | | Complex Sample Multivariate Multinomial logistic regression (Adjusted Odd Ratio, OR) | | | |
|---|---|---|---|---|---|---|---|---|
| | Single substance use vs Non-user | | Polysubstance use vs Non-user | | Single substance use vs Non-user | | Polysubstance use vs Non-user | |
| | Crude OR (95% CI) | p value | Crude OR (95% CI) | p value | Adjusted OR (95% CI) | p value | Adjusted OR (95% CI) | p value |
| **Sex** | | | | | | | | |
| Female | 1 | | 1 | | 1 | | 1 | |
| Male | 2.98 (2.59, 3.43) | <0.001 | 4.43 (3.39, 5.78) | <0.001 | 2.95 (2.57, 3.36) | <0.001 | 4.60 (3.59, 5.90) | <0.001 |
| **Age** | | | | | | | | |
| 16–17 years | 1 | | 1 | | 1 | | 1 | |
| 13–15 years | 1.36 (1.18, 1.56) | <0.001 | 0.87 (0.65, 1.16) | 0.349 | 1.45 (1.29, 1.64) | <0.001 | 1.07 (0.83, 1.37) | 0.605 |
| **Ethnic** | | | | | | | | |
| Malay | 1 | | 1 | | 1 | | 1 | |
| Chinese | 2.00 (1.71, 2.33) | <0.001 | 2.19 (1.58, 3.03) | <0.001 | 2.02 (1.72, 2.37) | <0.001 | 2.41 (1.75, 3.04) | <0.001 |
| Indian | 1.05 (0.88, 1.26) | <0.580 | 5.12 (3.34, 7.84) | <0.001 | 1.00 (0.76, 1.29) | 0.993 | 4.37 (2.92, 6.54) | <0.001 |
| Other Bumiputeras | 1.58 (1.30, 1.92) | <0.001 | 4.31 (3.12, 5.95) | <0.001 | 1.68 (1.36, 2.09) | <0.001 | 5.26 (3.63, 7.52) | <0.001 |
| **Locality of School** | | | | | | | | |
| Urban | 1 | | 1 | | | | | |
| Rural | 0.99 (0.86, 1.15) | 0.918 | 1.42 (0.96, 2.09) | 0.075 | - | - | - | - |
| **Marital Status of Parents** | | | | | | | | |
| Married and living together | 1 | | 1 | | 1 | | 1 | |
| Divorced/Widowed/Separated | 1.26 (1.11, 1.41) | <0.001 | 1.87 (1.54, 2.28) | <0.001 | 1.21 (1.06, 1.37) | 0.004 | 1.43 (1.13, 1.82) | 0.003 |
| **Truancy in the past 30 days** | | | | | | | | |
| No | 1 | | 1 | | 1 | | 1 | |
| Yes | 1.91 (1.73, 2.11) | <0.001 | 4.52 (3.80, 5.38) | <0.001 | 1.74 (1.57, 1.93) | <0.001 | 3.60 (3.01, 4.30) | <0.001 |
| **Peer support** | | | | | | | | |
| Adequate | 1 | | 1 | | 1 | | 1 | |
| Inadequate | 1.70 (1.55, 1.86) | <0.001 | 2.00 (1.72, 2.33) | <0.001 | 1.25 (1.14, 1.38) | <0.001 | 1.36 (1.13, 1.68) | 0.002 |
| **Parental or guardian bonding** | | | | | | | | |
| Adequate | 1 | | 1 | | 1 | | 1 | |
| Inadequate | 1.65 (1.52, 1.77) | <0.001 | 2.19 (1.88, 2.54) | <0.001 | 1.43 (1.31, 1.56) | <0.001 | 1.83 (1.50, 2.23) | <0.001 |
| **Parental or guardian never/ rarely respect for privacy** | | | | | | | | |
| Yes | 1 | | 1 | | | | | |
| No | 1.19 (1.07, 1.32) | 0.001 | 1.99 (1.70, 2.33) | <0.001 | - | - | - | - |
| **Anxiety** | | | | | | | | |
| No | 1 | | 1 | | | | | |
| Yes | 1.15 (1.05, 1.26) | 0.002 | 2.95 (2.36, 3.68) | <0.001 | - | - | - | - |
| **Stress** | | | | | | | | |
| No | 1 | | 1 | | | | | |
| Yes | 1.25 (1.08, 1.45) | 0.003 | 3.24 (2.61, 4.03) | <0.001 | - | - | - | - |
| **Depression** | | | | | | | | |
| No | 1 | | 1 | | 1 | | 1 | |
| Yes | 1.61 (1.44, 1.79) | <0.001 | 5.37 (4.09, 7.07) | <0.001 | 1.41 (1.31, 1.56) | <0.001 | 3.19 (2.55, 3.98) | <0.001 |
| **Loneliness** | | | | | | | | |
| No | 1 | | 1 | | | | | |
| Yes | 1.08 (0.94, 1.24) | 0.282 | 2.32 (1.88, 2.86) | <0.001 | - | - | - | - |
| **Had Close friend** | | | | | | | | |

*(Continued)*

**Table 4.** (Continued)

| Predictive factors | Complex Sample Univariate Multinomial logistic regression (Crude Odd Ratio, OR) | | | | Complex Sample Multivariate Multinomial logistic regression (Adjusted Odd Ratio, OR) | | | |
|---|---|---|---|---|---|---|---|---|
| | Single substance use vs Non-user | | Polysubstance use vs Non-user | | Single substance use vs Non-user | | Polysubstance use vs Non-user | |
| | Crude OR (95% CI) | p value | Crude OR (95% CI) | p value | Adjusted OR (95% CI) | p value | Adjusted OR (95% CI) | p value |
| Yes | 1 | | 1 | | 1 | | 1 | |
| No | 1.71 (1.38, 2.12) | <0.001 | 4.85 (3.39, 6.94) | <0.001 | 1.33 (1.04, 1.72) | <0.001 | 2.73 (1.83, 4.06) | <0.001 |
| **Being bullied** | | | | | | | | |
| No | 1 | | 1 | | 1 | | 1 | |
| Yes | 1.59 (1.42, 1.79) | <0.001 | 6.07 (4.91, 7.50) | <0.001 | 1.33 (1.04, 1.72) | <0.001 | 3.44 (2.83, 4.18) | <0.001 |

*Multinomial Logistic Regression using complex sampling design was applied.

The Complex Sample Enter method was used for variable selection. Interactions were checked and not reported. Overall fit the model for each binary logit was checked accordingly: Correctly weighted classified table (logit function 1 = 82.1%, logit function 2 = 94.7%), Weighted Area under ROC curve (logit function 1 = 71.6%, logit function 2 = 87.4%). The model was considered fit based on the classification table and area under the ROC curve

Chinese (single user OR = 2.0, poly user OR = 2.4) as compared to Malay. Those who lived with divorce/widower/separated parents also show a higher likelihood of being single substance users and polysubstance users (single user OR = 1.2, poly user OR = 1.4) compared to those who lived with both parents (Table 4).

Other associated factors for single and polysubstance use include played truant (single user OR = 1.7, poly user OR = 3.6), being bullied (single user OR = 1.3, poly user OR = 3.4), lack of peer support (single user OR = 1.3, poly user OR = 1.4), inadequate parental bonding (single user OR = 1.4, poly user OR = 1.8), had depression (single user OR = 1.4, poly user OR = 3.2) and had no close friend (single user OR = 1.3, poly user OR = 2.7) (Table 4).

## Discussion

Substance use has been a matter of concern throughout the world, especially substance use among adolescents. Understanding the pattern of substance use among adolescents is vital to creating the most suitable prevention, intervention, and harm reduction strategies. The prevalence of polysubstance use in this study (5.1%) was lower compared to the United States (7.6%) [23]. Alcohol and tobacco were the most common pair of substances used. In the United States, the most frequently paired polysubstance uses were alcohol and marijuana [5]. In contrast, in the South-east Asia region [4] and other countries [24–27], smoking and consuming alcohol were common co-occurrence of substance use among adolescents. A systematic review of polysubstance use among adolescents shows that the most common polysubstance use was alcohol, tobacco, and marijuana [27].

Like other countries, male adolescents in Malaysia showed higher prevalence [4,24] and higher odds [1,28,29] for polysubstance use than females. Johnson et al. reported that males are prone to use substances as they had more curiosity and desire for adventure as these substances give them a feeling of excitement [30]. The peer influence factor and awareness of harmful substance use also contributed to the differences in substance use among gender [31]. However, it is contrary to the studies from Korea and European countries, which reported a similar prevalence of polysubstance use between males and females [32,33].

Higher age was frequently associated with polysubstance use [34–36]. However, in this study, age was only significant on single substance use but not for polysubstance use. We found that the younger age group (13–15 years) were more likely to engage in single substance

use and smoking was the commonest among them. Such finding could be due to economic factors as cigarettes are the cheapest substance and easiest to access. This finding is in line with studies by Lim et al. [37] and Hammod et al. [8], where Malaysian adolescents started smoking at the young age of 14 years old. The accessibility of substances plays a vital role in initiating a single substance among the younger generation. The majority of adolescents smokers reported that it was easy to get cigarettes in Malaysia [8], and most of them were able to buy cigarettes by themselves [8]. Early initiation of smoking among younger age is a massive concern as some studies indicating that cigarettes are a gateway to other substance use such as alcohol and illicit drug [38,39].

Some studies reported that ethnic background was associated with substance use [23,40,41]. In this study, the "other Bumiputeras" group, which included Indigenous people of Sabah and Sarawak had the highest likelihood of being a polysubstance user. Culture and practice of respective ethnic groups could have played a vital role in substance use among adolescents. For example, the "other Bumiputera" group, which includes indigenous people from Sabah and Sarawak, consumes alcohol during the festive and social gatherings as culture and social obligation [42]. In contrast, all Malaysian Muslims are forbidden from drinking alcohol, and it is also illegal to sell liquor to a Muslim in Malaysia. Similarly, a study from Netherland found low consumption of alcohol use among students from a Muslim background [43]. A study by Brown et al. shows religiosity has consistently been associated with a lower level of alcohol consumption and marijuana use among adolescents [44]. Further research should be conducted to investigate factors related to ethnic variation, religion and culture on a combination of substance use among the adolescent.

Truancy and being bullied increased the odds of an adolescent in Malaysia to become polysubstance users by three times. It is similar to previous studies in the United States and Taiwan [44–47]. According to Henry et al., truancy is associated with increased odds of substance use initiation and leads to substantial escalating of substance use [48]. Truant adolescents tend to skip school in pairs or groups with mutual delinquent peers, which could lead them to substance use. Unsupervised time spent with friends during truancy increased risk for substance use which could ultimately result in an escalation of substance use [49]. An adolescent who were being bullied often experience anxiety and social concern, which could lead them to use a substance to reduce their symptoms. According to Durand et al., students who were being bullied may use a cigarette or other substances to reduce anxiety and increase their social image among peers [50]. However, further research is needed to fully understand why truancy and being bullied increase the risk of polysubstance use.

The epidemiological and clinical studies reported that substance use disorder is often associated with depression and vice versa [51,52]. This study suggested that having depression increased the odds of polysubstance use among students which is consistent with findings in the United States by Maslowsky et al. and Kelder et al., [53,54]. Depressed adolescents may "self-medicate" with tobacco, alcohol, or drugs to reduce their depression or other psychological stress. However, our finding warrants continued research to understand the nature of causal possibility between depression and substance, which may affect the intervention method for substance use among adolescents.

This study also found that a lack of protective factors such as peer support, parental bonding, and close friends increased odd of polysubstance use. An adolescent with these protective factors tends to have low risk-taking behaviour, high social bonding and gets involved with conventional peers [44]. Our study also shows that adolescents belonging to divorced or separated parents had increased odd of polysubstance use. Similar to the research done by Arkes et al., adolescents with divorced parents tend to use alcohol and marijuana [17].

## Conclusion

In summary, this study found many significant factors which were associated with substance use among adolescent. These findings could have substantial implications for its prevention strategies. Targeting high-risk groups is crucial for developing effective prevention and intervention strategies. School programs to create awareness on the dangers of substance use is essential and need to be targeted at males and younger age groups. According to Tan et al. existing drug education program in Malaysia such as *Tunas*, PINTAR (*Program Intelek Asuhan Rohani*) and *PIP* (*Program Intervensi Pelajar*) need to be revised as the number of substance use among adolescent in Malaysia keeps increasing over the years [55].

The gateway drug theory suggests that using one "softer" substance, such as tobacco and alcohol, can lead to the use of another "harder" substance, such as illicit drugs. Much attention needs to be given to prevent any "soft" substance initiation, especially at a young age. Tobacco, alcohol and illicit drugs control programs should focus on restricting adolescent access to such substances. Some recommended measures include increased fines and even revoking the license of those found to be selling these products to underage individuals. Increasing tobacco duty tax, creating effective media campaigns, and formulating comprehensive strategies on specific prevention and cessation programs are equally essential. Various innovative efforts need to be taken to prevent adolescents from any substance use as it has no proven benefits in the short or long term but only known to be detrimental in many ways.

## Strengths and limitations

This study has many strengths. It is a large, nationally representative sample of Malaysia. Also, It added value for the prevalence of polysubstance use in Malaysia by using correct analysis according to survey design weight. Besides, this study applied robust statistical analysis (multinomial logistic regression) which can differentiate between single substance users versus polysubstance users compared to non-user.

Despite its strength, this study also had its limitations. Firstly, this study used the data from a cross-sectional study design, which limits the ability to estimate the causal effect between predictive factors and outcome (substance use). The longitudinal cohort study is the best method in future research to examine the relation among these predictive factors. Also, the data was based on self–reporting, which may result in some bias results. However, the questions on substance use were self-completed by the respondent to respect their privacy and confidentiality, which also reduces some of the bias effects.

## Acknowledgments

We would like to thank the Director-General of Health Malaysia for his permission to publish this article.

## Author Contributions

**Conceptualization:** Wan Shakira Rodzlan Hasani, Thamil Arasu Saminathan, Halizah Mat Rifin.

**Data curation:** Wan Shakira Rodzlan Hasani.

**Formal analysis:** Wan Shakira Rodzlan Hasani, Hamizatul Akmal Abd Hamid.

**Investigation:** Wan Shakira Rodzlan Hasani.

**Methodology:** Wan Shakira Rodzlan Hasani.

**Software:** Wan Shakira Rodzlan Hasani.

**Validation:** Wan Shakira Rodzlan Hasani.

**Writing – original draft:** Wan Shakira Rodzlan Hasani, Thamil Arasu Saminathan, Nur Liana Ab Majid, Jane Ling Miaw Yn, Halizah Mat Rifin, Tania Gayle Robert Lourdes.

**Writing – review & editing:** Wan Shakira Rodzlan Hasani, Thamil Arasu Saminathan, Nur Liana Ab Majid, Jane Ling Miaw Yn, Halizah Mat Rifin, Hamizatul Akmal Abd Hamid, Tania Gayle Robert Lourdes, Ahzairin Ahmad, Hasimah lsmail, Rusdi Abd Rashid, Muhammad Fadhli Mohd Yusoff.

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
