## [Decision Letter · Decision Letter 0]

12 May 2020

PONE-D-20-05410

Single and polysubstance use among Adolescents in Malaysia: Finding from the National Health and Morbidity Survey 2017

PLOS ONE

Dear Mrs Rodzlan Hasani,

Thank you for submitting your manuscript to PLOS ONE. After careful consideration, we feel that it has merit but does not fully meet PLOS ONE’s publication criteria as it currently stands. Therefore, we invite you to submit a revised version of the manuscript that addresses the points raised during the review process.

We would appreciate receiving your revised manuscript by Jun 26 2020 11:59PM. To enhance the reproducibility of your results, we recommend that if applicable you deposit your laboratory protocols in protocols.io, where a protocol can be assigned its own identifier (DOI) such that it can be cited independently in the future. For instructions see: http://journals.plos.org/plosone/s/submission-guidelines#loc-laboratory-protocols

We look forward to receiving your revised manuscript.

Kind regards,

Samantha S. Goldfarb, DrPH

Academic Editor

PLOS ONE

Journal Requirements:

2. Please provide additional details regarding participant consent. In the ethics statement in the Methods and online submission information, please ensure that you have specified (1) whether consent was suitably informed and (2) what type you obtained (for instance, written or verbal). Since your study included minors under age 18, state whether you obtained consent from parents or guardians. If the need for consent was waived by the ethics committee, please include this information.

3. We noticed you have some minor occurrence(s) of overlapping text with the following previous publication(s), which needs to be addressed:

https://doi.org/10.1186/1471-2458-14-1044

https://doi.org/10.1177%2F1010539519865053

https://doi.org/10.1371/journal.pone.0207472

In your revision ensure you cite all your sources (including your own works), and quote or rephrase any duplicated text outside the Methods section. Further consideration is dependent on these concerns being addressed.

Additional Editor Comments (if provided):

Thank you for your submission. Please make the revisions suggested by Reviewer 1 and we will reconsider your manuscript for publication.

Reviewers' comments:

Reviewer's Responses to Questions

**Comments to the Author**

1. Is the manuscript technically sound, and do the data support the conclusions?

Reviewer #1: Partly

2. Has the statistical analysis been performed appropriately and rigorously? 

Reviewer #1: No

3. Have the authors made all data underlying the findings in their manuscript fully available?

Reviewer #1: Yes

4. Is the manuscript presented in an intelligible fashion and written in standard English?

Reviewer #1: No

5. Review Comments to the Author

Reviewer #1: Overall, the manuscript examined a topic that is important, timely, and appropriate for the PLOs One. This paper addresses an important topic, which is currently underexplored, in particular, in Southeast Asia. Thus, this is a welcome addition to what we know about the substance use among adolescents. However, I have the following queries/comments.

1. Introduction: What is the impacts of polysubstance use and why is such study important in prevention and treatment of substance users? The information would be mentioned in introduction to help readers to better understand the importance of your study.

In addition, on what basis did you choose those variables/risk factors? Also, in your introduction part, there was a very little usage of literature conducting in Malaysia and the neighboring countries on the similar event.

2. Methods: The authors have not discussed how they built the multivariate regression analysis model they are referring in Table 3 and 4 is not enough.

In page 13, you described “A final model was obtained which included all the factors that were significantly associated at the level of p <0.05.” you need to describe in detail. For example, table 3 presents crude odds ratios using univariate multinomial logistic regression. In cases of some variables, such as parental or guardian never/rarely respect for privacy, anxiety, and stress, showed statistically significances in table 4. but they were not included in the multivariate model. How did you build the multivariate regression analysis?

3. Results and discussion: Polysubstance use is defined as the use of more than one non-prescribed licit or illicit substance either concurrently or simultaneously. As it is difficult to understand the table 2, please revise the table readable.

In addition, the estimate in page 13 and 14 (5.1%) is different from that in page 17 (0.5%). I may assume that 0.5% is the prevalence of those who are concurrent use of three substances (alcohol, tobacco and illicit drug)? Please check and clarify the definition respectively.

4. Discussion: The results and discussion are lacking the nuts and bolts to would make the paper relevant to policy and practice.

In addition, in Table 2, other ethnicity compared to Malay, such as Chinese, Indian and others had higher odds of being polysubstance user. In particular, Indians and other ethnicities had much higher odds. However, the authors only mentioned Chinese and others and further failed to explain why it happened among Indians and others rather than Malay.

The authors mentioned that “the current study is the first study to examine self-reported substance use among a nationally representative sample of adolescents to better estimate the prevalence of polysubstance use among Malaysian adolescents.” To my knowledge, there are some studies on polysubstance use, although some evaluated concurrent use of alcohol and tobacco and not include illicit drug use using GSHS. So you may think of what your study can add in the literature.

5. Presentations of tables: I am troubled by table presentation style in Table 3 and 4. For example, in table 3, the estimates of single substance use are not in the same row. Need to revise table presentations properly in table 3 and 4.

Table 3: It would be better if you change some labels.

Polysubstance use of -> Polysubstance use (delete “of”)

Total -> Any substance use

Add one category of “Alcohol and tobacco and illicit drug” and the estimate.

6. Reference format: Please check the appropriate reference format of PlOS One and revise accordingly.

6. PLOS authors have the option to publish the peer review history of their article (what does this mean?). If published, this will include your full peer review and any attached files.

Reviewer #1: No

---

## [Author Response · Author response to Decision Letter 0]

24 Aug 2020

1. Introduction. What is the impacts of polysubstance use and why is such study important in prevention and treatment of substance users? The information would be mentioned in introduction to help readers to better understand the importance of your study. In addition, on what basis did you choose those variables/risk factors? Also, in your introduction part, there was a very little usage of literature conducting in Malaysia and the neighboring countries on the similar event.

Feedback: Thank you for your suggestion. We added the impacts of polysubstance use, factor related to substance use and prevalence of combination substance in Malaysian and neighboring countries (South-east Asia). Almost all paragraphs in introduction part were changed accordingly in order to help reader understand the important of this study. 

2. Method: The authors have not discussed how they built the multivariate regression analysis model they are referring in Table 3 and 4 is not enough.

In page 13, you described “A final model was obtained which included all the factors that were significantly associated at the level of p <0.05.” you need to describe in detail. For example, table 3 presents crude odds ratios using univariate multinomial logistic regression. In cases of some variables, such as parental or guardian never/rarely respect for privacy, anxiety, and stress, showed statistically significances in table 4. but they were not included in the multivariate model. How did you build the multivariate regression analysis?

Feedback: Thank you for your suggestion. This statement (below) on how the multivariate regression model was build were added under method section. “Univariable analysis was carried out in this study by testing all the 15 potential predictor variables in order to screen for important independent variables. Considering the variables with p-values <0.25 from univariable analysis and important variables based on biological plausibility, all those 15 variables were included in the preliminary final model (variable selection). The variable selection is the process of “reducing the model” in order to get the best fit model by including all the candidate variable in the model, and repeatedly removing the variables with the highest non-significant p value until the model contains only significant terms. Hence, the final model was created based on ten variables that were significantly associated at the level of p <0.05 during final steps of variables selection. Those variables were age, sex, and ethnicity, marital status of parents, truancy, peer support, parent/guardian bonding, depression, close friend and being bullied”

3. Results: Polysubstance use is defined as the use of more than one non-prescribed licit or illicit substance either concurrently or simultaneously. As it is difficult to understand the table 2, please revise the table readable.

In addition, the estimate in page 13 and 14 (5.1%) is different from that in page 17 (0.5%). I may assume that 0.5% is the prevalence of those who are concurrent use of three substances (alcohol, tobacco and illicit drug)? Please check and clarify the definition respectively.

Feedback: Thank you for your suggestion. For Table 2 – Revision done. 5.1% is the correct value. We removed the value 0.5% in page 17. 

4. Discussion: 

Comment: The results and discussion are lacking the nuts and bolts to would make the paper relevant to policy and practice.

Feedback: Thank you for opinion. Almost all paragraphs under discussion part were changed accordingly in order to make this paper relevant to policy and practice. The recommendation specific to policy & practice were added under conclusion.

Comment: In addition, in Table 2, other ethnicity compared to Malay, such as Chinese, Indian and others had higher odds of being polysubstance user. In particular, Indians and other ethnicities had much higher odds. However, the authors only mentioned Chinese and others and further failed to explain why it happened among Indians and others rather than Malay.

Feedback: Majority of other ethnicities were Bumiputeras Sabah & Sarawak. Thus the word “other” was changed to “Other Bumiputeras”. The explanation why ethnic groups other than Malay more likely became polysusbtance user were explained details in discussion.

Comment: The authors mentioned that “the current study is the first study to examine self-reported substance use among a nationally representative sample of adolescents to better estimate the prevalence of polysubstance use among Malaysian adolescents.” To my knowledge, there are some studies on polysubstance use, although some evaluated concurrent use of alcohol and tobacco and not include illicit drug use using GSHS. So you may think of what your study can add in the literature.

Feedback: Thanks you for your comment. The statement “the current study is the first study to examine self-reported substance use among a nationally representative sample of adolescents to better estimate the prevalence of polysubstance use among Malaysian adolescents” was removed. 

5. Presentations of tables. 

Comment: I am troubled by table presentation style in Table 3 and 4. For example, in table 3, the estimates of single substance use are not in the same row. Need to revise table presentations properly in table 3 and 4.

Feedback: Thank you for your suggestion. Done editing Table 3 & 4. 

Comment: Table 3: It would be better if you change some labels. Polysubstance use of -> Polysubstance use (delete “of”)

Total -> Any substance use.

Feedback: Done editing label. Thank you for your suggestion. 

Comment: Add one category of “Alcohol and tobacco and illicit drug” and the estimate.

Feedback: Analysis for use all three substance (alcohol+ tobacco + drug) was done and added to Table 3. Thank you for the suggestion. 

6. Reference format:Please check the appropriate reference format of PlOS One and revise accordingly.

Feedback: Revision done based on PLOS one format. Thank you for your comment.

---

## [Decision Letter · Decision Letter 1]

14 Oct 2020

PONE-D-20-05410R1

Polysubstance use among Adolescents in Malaysia: Finding from the National Health and Morbidity Survey 2017

PLOS ONE

Dear Dr. Rodzlan Hasani,

Thank you for submitting your manuscript to PLOS ONE. After careful consideration, we feel that it has merit but does not fully meet PLOS ONE’s publication criteria as it currently stands. Therefore, we invite you to submit a revised version of the manuscript that addresses the points raised during the review process.

We look forward to receiving your revised manuscript.

Kind regards,

Michelle Tye, Ph.D.

Academic Editor

PLOS ONE

Additional Editor Comments (if provided):

As per Reviewer 2's comment, this manuscript requires significant English language editing to enhance readability.

Some additional information in the Methods section as to how consent for participation in the survey was sought and received would be helpful, as well as the duration of the survey. Was it undertaken during school time, or outside of school hours? Though the authors do note that the methods are reported elsewhere, the key methodological details should still be reported in this manuscript.

Reviewers' comments:

Reviewer's Responses to Questions

**Comments to the Author**

1. If the authors have adequately addressed your comments raised in a previous round of review and you feel that this manuscript is now acceptable for publication, you may indicate that here to bypass the “Comments to the Author” section, enter your conflict of interest statement in the “Confidential to Editor” section, and submit your "Accept" recommendation.

Reviewer #1: (No Response)

Reviewer #2: (No Response)

2. Is the manuscript technically sound, and do the data support the conclusions?

Reviewer #1: Partly

Reviewer #2: Partly

3. Has the statistical analysis been performed appropriately and rigorously? 

Reviewer #1: I Don't Know

Reviewer #2: Yes

4. Have the authors made all data underlying the findings in their manuscript fully available?

Reviewer #1: No

Reviewer #2: Yes

5. Is the manuscript presented in an intelligible fashion and written in standard English?

Reviewer #1: No

Reviewer #2: No

6. Review Comments to the Author

Reviewer #1: Thanks for the revision.

First, Introduction: the authors made a revision. Not agree to add the sentences about patterns of polysubstance use in the USA in the first paragraph of the introduction; “In the United States, the most frequently paired of polysubstance use …….both alone and in combination.” you may move to discussion part to discuss the polysubstance use patterns among countries.

Second, the authors added the new variables of "all three substance use(alcohol+ tobacco + drug)." It is surprising that prevalence of alcohol+ tobacco + drug is higher than that of the other polysubstance user group. For example, the prevalence of alcohol+ tobacco + drug (2.2%) is higher compared to that of tobacco and alcohol use (2.0%).

i) why is the prevalence of alcohol+ tobacco + drug higher than that of tobacco and alcohol use (2.0%)? As illicit substance is not common, it seems interesting and also doubtable. Please discuss in discussion part.

ii) in the first paragraph of the discussion, the authors described “Overall, this study showed alcohol and tobacco were the most common pair of substance use.” please clarify the sentences and the estimates in Table 3.

Lastly, regarding the reference formatting, the references should be revised correctly. In particular, the name list of the authors were not well-formatted.

Reviewer #2: This article examines the prevalence of and associated factors related to single-use and polysubstance use among adolescents in Malaysia. I would like to point out that I am reviewing a revised draft of the manuscript (I did not review the initial draft); thus, my comments are relatively minor since the authors seem to have addressed many of the previous reviewers’ concerns. Overall, the subject matter is of great public health importance and the authors have attempted to present a comprehensive profile of juvenile substance use in Malaysia.

1. The introduction, while succinct, can benefit from a slightly more enhanced discussion of the topic at hand. Specifically, the following can be included:

a. Importance of identifying and/or preventing polysubstance use at an early stage.

b. Ease of access to alcohol and drugs within the context of Malaysia, that can explain the rising rates of polysubstance use.

2. I’m a little concerned about the self-reported nature of the truancy variable. Was it not possible to retrieve this information from classroom records? Similarly, the use of single items for peer support, parental bonding and parental respect for privacy, is short-sighted and unreliable.

3. The writing is poor and idiosyncratic. There are multiple instances of sentences that are poorly worded and confusing to read. For example, “polysubstance use between the male and female” sounds odd. This sentence should be reworded as “polysubstance use between males and females. Consider this other statement “For example, Indigenous people (Sabah and Sarawak) is required to drink alcohol during harvest festival and social gathering [43] but Malays where mostly are Muslims are prohibited from drink alcohol according to Malaysia’s sharia law.” I realize that there is likely a language barrier and hence I would encourage the authors to seek the assistance of a copy-writer to make the necessary corrections.

4. One of the most interesting findings that the authors gloss over is the that younger adolescents were more likely to engage in single substance use. How do the authors explain this result? As I mentioned in my first comment, it would be useful to understand the socio-cultural context that either constrains or emboldens substance use at a young age. This would allow for policy or structural changes to address these issues.

7. PLOS authors have the option to publish the peer review history of their article (what does this mean?). If published, this will include your full peer review and any attached files.

Reviewer #1: No

Reviewer #2: No

---

## [Author Response · Author response to Decision Letter 1]

27 Nov 2020

Response to academic editor comments

1. Additional Editor Comments (if provided):

As per Reviewer 2's comment, this manuscript requires significant English language editing to enhance readability.

Some additional information in the Methods section as to how consent for participation in the survey was sought and received would be helpful, as well as the duration of the survey. Was it undertaken during school time, or outside of school hours? Though the authors do note that the methods are reported elsewhere, the key methodological details should still be reported in this manuscript.

Feedback: Thank you for your comments. The English language editing was done accordingly. The statement regarding consent for participants already stated in the methodology under ethical approval as below;

“Prior to the survey, several meetings with person in charge at selected schools were conducted to inform them about the survey and brief about the parents’ consent form. The consent forms were distributed a week prior the survey. Students and their parents’ consent forms were collected just prior to the actual day of survey. Student who did not receive parental or guardian consent or they themselves refused to participate were considered as non-response of eligible participants in this survey”.

The duration of the survey and field implementation was added in the methodology as below:

“Prior to the survey, the data collectors were trained during 1-week workshop. The data collection was conducted from March until May 2017. A field supervisor was assigned for each state to supervise the survey activities including arrangement the slot for survey with the school. The survey was conducted either in the classroom or school hall during school time and it took about 45 minutes to 1 hour to complete the survey. The answered OMR form was put in the envelope by each student to maintain confidentiality. ” 

Response to reviewer comments

2. Review Comments to the Author

Reviewer #1: Thanks for the revision.

First, Introduction: the authors made a revision. Not agree to add the sentences about patterns of polysubstance use in the USA in the first paragraph of the introduction; “In the United States, the most frequently paired of polysubstance use …….both alone and in combination.” you may move to discussion part to discuss the polysubstance use patterns among countries.

Feedback: Thank you for your suggestion. We moved this statement to discussion.

Second, the authors added the new variables of "all three substance use(alcohol+ tobacco + drug)." It is surprising that prevalence of alcohol+ tobacco + drug is higher than that of the other polysubstance user group. For example, the prevalence of alcohol+ tobacco + drug (2.2%) is higher compared to that of tobacco and alcohol use (2.0%).

i) why is the prevalence of alcohol+ tobacco + drug higher than that of tobacco and alcohol use (2.0%)? As illicit substance is not common, it seems interesting and also doubtable. Please discuss in discussion part.

Feedback: Thank you for your comment. We agree with your point. After discuss with other co-authors, we re-analyse the prevalence of combination two substances regardless of the 3rd substance use (Refer to Table 3). Previously, we exclude the third substance on combination of two substance (example combination of alcohol and tobacco excluded the drug user, combination of drug and alcohol excluded tobacco user). 

The illustration circle of combination substance use (in the previous analysis) as below; 

After re-analyse, the prevalence of combination three substances (2.2%) is lower than combination of any two substances. The illustration as below

We agree that, the current result for combination of substance is more meaningful and will interpret the result as current analysis. 

ii) in the first paragraph of the discussion, the authors described “Overall, this study showed alcohol and tobacco were the most common pair of substance use.” please clarify the sentences and the estimates in Table 3.

Feedback: After re-analyse for Table 3, the combination of alcohol and tobacco use shows the highest prevalence as compare to combination of other substance. Thus this statement (“Overall, this study showed alcohol and tobacco were the most common pair of substance use”) will be maintain.

Lastly, regarding the reference formatting, the references should be revised correctly. In particular, the name list of the authors were not well-formatted.

Feedback: Thank you for your comment. The references was revised accordingly. We change to PLoS reference style using EndNote. 

3. Reviewer #2: 

This article examines the prevalence of and associated factors related to single-use and polysubstance use among adolescents in Malaysia. I would like to point out that I am reviewing a revised draft of the manuscript (I did not review the initial draft); thus, my comments are relatively minor since the authors seem to have addressed many of the previous reviewers’ concerns. Overall, the subject matter is of great public health importance and the authors have attempted to present a comprehensive profile of juvenile substance use in Malaysia.

1. The introduction, while succinct, can benefit from a slightly more enhanced discussion of the topic at hand. Specifically, the following can be included:

a. Importance of identifying and/or preventing polysubstance use at an early stage.

b. Ease of access to alcohol and drugs within the context of Malaysia, that can explain the rising rates of polysubstance use.

Feedback: Thank you for your suggestion. We revised the introduction and added the statement as suggested. 

2. I’m a little concerned about the self-reported nature of the truancy variable. Was it not possible to retrieve this information from classroom records? Similarly, the use of single items for peer support, parental bonding and parental respect for privacy, is short-sighted and unreliable.

Feedback: Thank you for your comment. As described in method section, this study applied self-administered method for data collection using validated questionnaire (GSHS) from WHO. We followed all the protocols and guideline from WHO for GSHS. Brief information regarding GSHS and link (URL) for GSHS as below;

“The Global School-based Student Health Survey (GSHS) was developed by the World Health Organization (WHO) and the Centers for Disease Control and Prevention (CDC). It has been widely use by more than 100 countries, which enable international comparisons of findings. 

The GSHS is a school-based survey conducted primarily among students aged 13–17 years. The GSHS uses a standardized scientific sample selection process; common school-based methodology; and core questionnaire modules, core-expanded questions, and country-specific questions that are combined to form a self-administered questionnaire that can be administered during one regular class period. 

The 10 core questionnaire modules address the leading causes of morbidity and mortality among children and adults worldwide which were 1) Alcohol use, 2)Dietary behaviors, 3)Drug use, 4) Hygiene, 5) Mental health, 6) Physical activity, 7) Protective factors, 8) Sexual behaviors that contribute to HIV infection, other sexually-transmitted infections, and unintended pregnancy , 9)Tobacco use, 10) Violence and unintentional injury”

https://www.cdc.gov/gshs/background/index.htm

https://www.cdc.gov/gshs/pdf/GSHSOVerview.pdf

In the other hand, this study involved 212 selected schools with 27, 497 sample students collected in year 2017. We are unable to retrieve the information on classroom report for truancy as we did not collect that information during data collection. 

3. The writing is poor and idiosyncratic. There are multiple instances of sentences that are poorly worded and confusing to read. For example, “polysubstance use between the male and female” sounds odd. This sentence should be reworded as “polysubstance use between males and females. Consider this other statement “For example, Indigenous people (Sabah and Sarawak) is required to drink alcohol during harvest festival and social gathering [43] but Malays where mostly are Muslims are prohibited from drink alcohol according to Malaysia’s sharia law.” I realize that there is likely a language barrier and hence I would encourage the authors to seek the assistance of a copy-writer to make the necessary corrections.

Feedback: Thank you for your comment and suggestion. The sentences were rephrased and editorial correction was done and revised accordingly. 

4. One of the most interesting findings that the authors gloss over is the that younger adolescents were more likely to engage in single substance use. How do the authors explain this result? As I mentioned in my first comment, it would be useful to understand the socio-cultural context that either constrains or emboldens substance use at a young age. This would allow for policy or structural changes to address these issues.

Feedback: Thank you for your comment. We added the explanation on the younger adolescent more likely to engage in single substance use on the discussion.

---

## [Decision Letter · Decision Letter 2]

11 Dec 2020

PONE-D-20-05410R2

Polysubstance use among Adolescents in Malaysia: Finding from the National Health and Morbidity Survey 2017

PLOS ONE

Dear Dr. Rodzlan Hasani,

Thank you for submitting your manuscript to PLOS ONE. After careful consideration, we feel that it has merit but does not fully meet PLOS ONE’s publication criteria as it currently stands. Therefore, we invite you to submit a revised version of the manuscript that addresses the points raised during the review process.

Thank you for your thorough revisions, Reviewer 1 has noted some additional minor comments to address. 

We look forward to receiving your revised manuscript.

Kind regards,

Michelle Tye, Ph.D.

Academic Editor

PLOS ONE

Reviewers' comments:

Reviewer's Responses to Questions

**Comments to the Author**

1. If the authors have adequately addressed your comments raised in a previous round of review and you feel that this manuscript is now acceptable for publication, you may indicate that here to bypass the “Comments to the Author” section, enter your conflict of interest statement in the “Confidential to Editor” section, and submit your "Accept" recommendation.

Reviewer #1: All comments have been addressed

Reviewer #2: All comments have been addressed

2. Is the manuscript technically sound, and do the data support the conclusions?

Reviewer #1: Yes

Reviewer #2: Yes

3. Has the statistical analysis been performed appropriately and rigorously? 

Reviewer #1: No

Reviewer #2: Yes

4. Have the authors made all data underlying the findings in their manuscript fully available?

Reviewer #1: Yes

Reviewer #2: Yes

5. Is the manuscript presented in an intelligible fashion and written in standard English?

Reviewer #1: Yes

Reviewer #2: Yes

6. Review Comments to the Author

Reviewer #1: Thanks for your revision. I found the manuscript improved. I have more minor things to be revised though.

Abstract

“The NHMS utilised a two-stage stratified cluster sampling using.”

Tables

In table 2, the prevalence of the lifetime use of smoking tobacco is supposed to be greater than that of the current use. but in your analysis, the lifetime use (5.1%) vs the current use (6.7%). please check the data.

In Table 3, indicate in the table whether the estimates were crude or weighted values.

In Table 4, the 95% CI of the COR of Chinese need to be checked (COR=1.10 (95% CI=1.71-2.33)). The COR is supposed to be in the range of 95% CI.

Conclusion

Check the full name of “INTIM camp (Kem Kecemerlangan Intelek Murid)”

Reviewer #2: The authors have undertaken a thorough revision based on the comments made by other reviewers and myself. All my comments were addressed, the most significant being extensive copy-editing of the manuscript. I have no problem recommending that the manuscript be accepted for publication.

7. PLOS authors have the option to publish the peer review history of their article (what does this mean?). If published, this will include your full peer review and any attached files.

Reviewer #1: No

Reviewer #2: No

---

## [Author Response · Author response to Decision Letter 2]

30 Dec 2020

Response to reviewer comments

1. Abstract

“The NHMS utilised a two-stage stratified cluster sampling using.”

Feedback: Thank you for your comments. The word “using” was removed from the abstract. 

2. Tables

In table 2, the prevalence of the lifetime use of smoking tobacco is supposed to be greater than that of the current use. but in your analysis, the lifetime use (5.1%) vs the current use (6.7%). please check the data.

Feedback: Thank you for your comments. I do agree with you. After checked the analysis, I admit my mistake that I’m taking the wrong variable for lifetime use for smoked tobacco (the current figure demonstrated lifetime use for cigarette only). The value for lifetime use for smoked tobacco supposed be 21.4% (95%CI: 20.04, 22.84) for total, 34.3% (95%CI: 32.09, 36.55) for male and 8.7% (95%CI: 7.71, 9.88) for female. We really appreciate on your comment and apologies for the mistake. These values (in Table 2) were changed accordingly. 

In Table 3, indicate in the table whether the estimates were crude or weighted values.

Feedback: Thank you for your comment and suggestion. Table 2 and 3 are weighted analysis using complex sampling design analysis. The statement on weighted analysis added on the table’s foot note 

In Table 4, the 95% CI of the COR of Chinese need to be checked (COR=1.10 (95% CI=1.71-2.33)). The COR is supposed to be in the range of 95% CI.

Feedback: Thank you for your comments. I checked the output of the analysis. The value is wrongly entered. The COR of Chinese group should be 2.00, not 1.10. Thank you again for checking this table. 

3. Conclusion

Check the full name of “INTIM camp (Kem Kecemerlangan Intelek Murid)”

Feedback: Thank you for your comments. We checked from AADK/NADA website, INTIM or “Kem Kecemerlangan Intelek Murid” currently known as PINTAR or “Program Intelek Asuhan Rohani”. No full abbreviation for INTIM was found. Thus we decided to remove the word “INTIM (Kem Kecemerlangan Intelek Murid)” and changed to PINTAR (Program Intelek Asuhan Rohani). The details on PINTAR program as link below:

https://www.adk.gov.my/en/prevention/drug-free-education-institution/

---

## [Editor Report · Decision Letter 3]

5 Jan 2021

Polysubstance use among Adolescents in Malaysia: Finding from the National Health and Morbidity Survey 2017

PONE-D-20-05410R3

Dear Dr. Rodzlan Hasani,

We’re pleased to inform you that your manuscript has been judged scientifically suitable for publication and will be formally accepted for publication once it meets all outstanding technical requirements.

Kind regards,

Michelle Tye, Ph.D.

Academic Editor

PLOS ONE
---

## [Editor Report · Acceptance letter]

8 Jan 2021

PONE-D-20-05410R3 

Polysubstance use among adolescents in Malaysia: Findings from the National Health and Morbidity Survey 2017 

Dear Dr. Rodzlan Hasani:

I'm pleased to inform you that your manuscript has been deemed suitable for publication in PLOS ONE. Congratulations! Your manuscript is now with our production department. 

Kind regards, 

on behalf of

Dr. Michelle Tye 

Academic Editor

PLOS ONE